# Reference Genome Sequences of the Oriental Armyworm, *Mythimna separata* (Lepidoptera: Noctuidae)

**DOI:** 10.3390/insects13121172

**Published:** 2022-12-17

**Authors:** Kakeru Yokoi, Seiichi Furukawa, Rui Zhou, Akiya Jouraku, Hidemasa Bono

**Affiliations:** 1Insect Design Technology Group, Division of Insect Advanced Technology, Institute of Agrobiological Sciences, National Agriculture and Food Research Organization (NARO), 1-2 Owashi, Tsukuba 305-0901, Japan; joraku@affrc.go.jp; 2Institute of Life and Environmental Sciences, University of Tsukuba, Tsukuba 305-8572, Japan; furukawa.seiichi.ew@u.tsukuba.ac.jp; 3Degree Program in Agro-Bioresources Science and Technology, University of Tsukuba, Tsukuba 305-8572, Japan; syuuruizz@gmail.com; 4Laboratory of Genome Informatics, Graduate School of Integrated Sciences for Life, Hiroshima University, 3-10-23 Kagamiyama, Higashi-Hiroshima City 739-0046, Japan; bonohu@hiroshima-u.ac.jp; 5Laboratory of BioDX, Genome Editing Innovation Center, Hiroshima University, 3-10-23 Kagamiyama, Higashi-Hiroshima City 739-0046, Japan

**Keywords:** genome sequence, *Mythimna separata*, toll pathway, IMD pathway, C-type lectin

## Abstract

**Simple Summary:**

The oriental armyworm, *Mythimna separata*, is an agricultural pest that is commonly used to study insect immune reactions and interactions with parasitoid wasps. To promote such studies, a reference genome was constructed. The *M. separata* genome is 682 Mbp long, a size comparable to that of other lepidopteran insects. The contig N50 value of the genome is 2.7 Mb, which indicates sufficient quality to be used as reference genome data. Gene set data were constructed using genome and RNA-sequencing data, with a total of 21,970 genes and 24,452 coding sites being predicted. Functional gene annotation was performed using the predicted amino acid sequences and reference gene set data of the model organism and other insect species as well as the Unigene and Pfam datasets. Consequently, 45–80% of the amino acid sequences were annotated using these datasets. Using these data, most of the orthologs of core components in the Toll and immune deficiency (IMD) pathways were identified, suggesting the presence of these two pathways in *M. separata*. Additionally, 105 C-type lectins were identified in the *M. separata* genome, which were more numerous than those in other insect species, suggesting that these genes may be duplicated.

**Abstract:**

Lepidopteran insects are an important group of animals, including those used as biochemical and physiological model species in the insect and silk industries as well as others that are major agricultural pests. Therefore, the genome sequences of several lepidopteran insects have been reported. The oriental armyworm, *Mythimna separata*, is an agricultural pest commonly used to study insect immune reactions and interactions with parasitoid wasps as hosts. To improve our understanding of these research topics, reference genome sequences were constructed in the present study. Using long-read and short-read sequence data, de novo assembly and polishing were performed and haplotigs were purged. Subsequently, gene predictions and functional annotations were performed. To search for orthologs of the Toll and Immune Deficiency (IMD) pathways and for C-type lectins, annotation data analysis, BLASTp, and Hummer scans were performed. The *M. separata* genome is 682 Mbp; its contig N50 was 2.7 Mbp, with 21,970 genes and 24,452 coding sites predicted. All orthologs of the core components of the Toll and IMD pathways and 105 C-type lectins were identified. These results suggest that the genome data were of sufficient quality for use as reference genome data and could contribute to promoting *M. separata* and lepidopteran research at the molecular and genome levels.

## 1. Introduction

Several species of lepidopteran insects are known, of which there are both beneficial and detrimental examples. Although the domestic silkworm (*Bombyx mori;* Bombycidae; Linnaeus, 1758) is used for silk production, lepidopteran insects include major agricultural pests as well. For example, the fall armyworm (*Spodoptera frugiperda;* Noctuidae; J.E. Smith, 1797)*,* the beet armyworm *(Spodoptera exigua*; Noctuidae; Hübner, 1808)*,* the tobacco cutworm (*Spodoptera litura*; Noctuidae; Fabricius, 1775), the diamondback moth (*Plutella xylostella;* Plutellidae; Linnaeus, 1758), the cabbage looper (*Trichoplusia ni*; Noctuidae; Hübner, 1800–1803), and the oriental armyworm (*Mythimna separata;* Noctuidae; Walker, 1865) are among the most severe pests in certain crops [1,2,3]. In contrast, the tobacco hornworm (*Manduca sexta;* Sphingidae; Linnaeus, 1763) has been used as a biochemical and physiological model insect species (e.g., to study immune reaction, development, and metamorphosis [4,5]). Because of this importance research on the lepidopteran family of insects is quite active, and the genomes of lepidoptera species have been sequenced. The first lepidopteran genome sequence data to be reported was the draft genome sequence data of *Bombyx mori* in 2004 [6,7]. The genome data was later updated [8], and the chromosome-level genome sequence of *B. mori* was reported as well [9]. Additionally, genome sequence data of other lepidopteran families, including *S. frugiperda* [10,11], *S. exigua* [12], *S. litura* [2], *P. xylostella* [13,14], *T. ni* [15], and *M. sexta*, have been reported [16,17], and the genome sequence data for several butterflies were reported as well [18].

*M. separata* is commonly used to study immune reactions in insects. Insects exhibit humoral and cellular immune responses to foreign microorganisms [19]. *M. separata* larval hemocytes, especially granular cells and plasmatocytes, play a central role in cellular immune reactions; different types of invaders activate different reactions such as encapsulation, phagocytosis, and nodule formation. Several molecules in the *M. separata* hemolymph mediate these reactions. Growth-blocking peptides, found as insect cytokines, trigger the spreading of plasmatocytes [20]. C-type lectins (CTLs) are a superfamily of proteins that recognize carbohydrates in a calcium-dependent manner. Certain CTLs can activate cellular immune reactions. Ishihara et al. (2017) reported that the CTL encapsulation promoting lectin (EPL) enhances encapsulation [21], and another CTL, IML-10, promotes encapsulation as well [22]. A transcriptome analysis found 35 CTLs from *M. separata* larvae [23], many of which changed their expression profiles upon bacterial exposure, suggesting that they are involved in immune regulation. To study the interactions between *M. separata* and parasitic wasps or flies, the oriental armyworm can be used as a host insect for these species [24,25,26,27,28]. Using both species, our group identified several candidate factors needed for successful parasites in the braconid wasp *Meteorus pulchricornis* (Hymenoptera: Braconidae; Haliday, 1835) [29] and investigated several gene functions of *M. separata* related to apoptosis or immune reactions considered to be essential for parasitoids [30,31].

In order to deepen our understanding of the molecular mechanisms of immune reactions and interactions between *M. separata* and parasitoid wasps, we constructed a reference-quality genome sequence of the oriental armyworm (*M. separata*) using long-read and short-read sequence data. Consequently, gene set data and the translated amino acid sequence data were prepared and functional annotations of the predicted genes were performed (Figure 1). Using the gene set data, the orthologues consisting of IMD and Toll pathways, which are major immune signaling pathways, were both searched. Finally, CTL genes were searched in the gene sets and sequence and domain analysis of CTL were performed. These results suggest that the reference genome data and gene set data can contribute to promoting research on *M. separata* and Lepidopteran research at the molecular or genomic level.

## 2. Materials and Methods

### 2.1. Sample Preparation and Sequencing

*M. separata* were supplied from stock cultures stored at Takeda Chemical Industries, Ltd. (Osaka, Japan) [32] and maintained in the Laboratory of Applied Entomology and Zoology, University of Tsukuba, Japan. The insects were reared on an artificial diet (Silkmate, Nihon Nosan Kogyo, Kanagawa, Japan) at 25 ± 2 °C, 40–80% relative humidity (r. h.), and an L16:D8 photoperiod.

Genomic DNA from single male pupae was extracted using NucleoBond HMW DNA (TaKaRa Bio Inc, Shiga, Japan) according to manufacturer’s protocol. Briefly, the whole body in 500 µL Lysis buffer was homogenized with a plastic pestle and liquid nitrogen. After treatments of 200 µL Liquid Proteinase K at 50 °C for 2 h and 100 µL Liquid RNase A at 25 °C for 5 min, the homogenate was loaded onto a NucleoBond HMW Column. Through a washing step, DNA was finally eluted with 5 µL Elution buffer (Prep Kit 2.0, Pacific Bioscience, CA, USA) according to manufacturer’s protocol. The sequencing library was size-selected using the BluePippin system (Saga Science, Tokyo, Japan) with a lower cutoff of 30 kbp. One SMRT Cell 8M was sequenced on the PacBio Sequel II System with Binding Kit 2.0 and Sequencing Kit 2.0, yielding a total of 6,971,329 polymerase reads (201,369,397,241 bp) (Shearing System M220, Covaris Inc., Woburn, MA, USA). A paired-end library was constructed with a TruSeq DNA PCR-Free Library Prep kit (Illumina, San Diego CA, USA) and was size-selected on an agarose gel using a Zymoclean Large Fragment DNA Recovery Kit (Zymo Research, Orange, CA, USA). The final library was sequenced on the Illumina NovaSeq 6000 sequencer with a read length of 150 bp.

Total RNA samples were prepared from the hemocytes of *M. separata* larvae using TRIzol (Thermo Fisher Scientific, Waltham, MA, USA). PolyA RNA libraries were constructed using the TruSeq Stranded mRNA Library Prep Kit (Illumina), and sequencing (paired-end sequencing with 100-nt reads) was performed on a NovaSeq6000 platform (Illumina).

All raw sequence data were deposited in the Sequence Read Archive (SRA) of the DNA Data Bank of Japan (DDBJ). SRA accession IDs of the raw sequence data used in this study are listed in Appendix A.

### 2.2. Genome Assembly and Gene Prediction

The output BAM file of Sequel II was converted to a FASTA file using BAM2fastx version 1.3.1 (URL: https://github.com/PacificBiosciences/bam2fastx, accessed on 2 September 2022). De novo assembly was performed using the converted FASTA file with Flye version 2.9-b1774 with default settings [33]. Polishing of the assembled genome was performed using Minimap2 version 2.17 (mapping) [34], SAMtools version 1.10 (file conversion) [35], and Plion version 1.23 (perform polishing) [36]. Identification of the allelic contig pairings and production haplotype-fused assemblies were performed using Purge Haplotigs version 1.1.2 (URL: https://bitbucket.org/mroachawri/purge_haplotigs/src/master/) (accessed on 2 September 2022 ). The genome sequences were assessed using BUSCO version 5.2.2 with eukaryota_odb10 file [37]. The status of the genome sequences was evaluated using assembly-stats version 1.0.1 (URL: https://github.com/sanger-pathogens/assembly-stats accessed on 2 September 2022). The finished genome sequences were deposited to DDBJ/ENA/GenBank (accession IDs: BSAF01000001-BSAF01000569; Appendix A).

To search for repetitive sequences and transposable elements (TEs), RepeatModer2 version DEV and RepeatMasker version 4.1.2-p1 were used [38,39]. Gene prediction was performed using masked sequences and RNA sequencing (RNA-Seq) data, including those from the public database, as hint data (Appendix A). RNA-Seq data mapping to the reference genome was performed using HISAT2 version 2.2.1 [40], and mapped data were converted using SAMtools. Gene predictions were performed twice using Braker2 version 2.1.6 with RNA-Seq data hint-only mode and protein data hint-only mode independently [41]. The data output from RNA-Seq data hint-only mode and protein data hint-only mode were merged by TSEBRA version 1.0.3 [42]. The amino acid sequences of the gene sets were functionally annotated by the functional annotation workflow Fanflow4Insects [43]. Functional annotation included assignment of the top hit genes from comprehensively annotated organisms using the sequence similarity search program in global alignment (GGSEARCH) (https://fasta.bioch.virginia.edu/; accessed on 1 August 2022) and in the protein domains using the protein domain database Pfam (http://pfam.xfam.org/; accessed on 1 August 2022) via HMMSCAN in the HMMER package (http://hmmer.org/; accessed on 1 August 2022). The accession IDs of the raw read sequences of for RNA-Seq data are listed in Appendix A. To remove contaminating microbe sequences, the predicted amino acid sequences were annotated by BLAST search (version 2.12.0+) using the BLAST nr database (downloaded via the following URL: https://ftp.ncbi.nlm.nih.gov/blast/db/ accessed on 3 March 2022) containing 444,154,253 protein sequences.

### 2.3. Search for and Sequence Analysis of Immune-Related Genes in M. separata

To search for genes related to the IMD and Toll pathways, a BLASTx or BLASTp (version 2.2.31+) search was performed using *Drosophila melanogaster* (Diptera; Meigen, 1830) fruit fly sequences as query sequences from FlyBase (URL: https://flybase.org/ accessed on 1 October 2022) and the amino acid sequences of *M. separata* were predicted as subject sequences. The phylogenetic tree construction (Neighbor Joining (NJ) method with Jacard methods) and subsequent domain analysis of peptidoglycan recognition proteins (PGRPs) and Gram-negative binding protein (GNBP) (hmmscan version 3.1b2 with Pfam-A.hmm of Pfam_35) were performed by DoMosaics (URL: https://domainworld.uni-muenster.de/developing/domosaics/version 0.95 accessed on 1 October 2022) [44], while other phylogenetic trees of PGRPs and GNBPs (Maximum Likelihood method (ML)) were constructed by MEGA X with default settings [45]. Phylogenetic tree construction (Neighbor joining method) and domain analysis of CTLs were performed by DoMosaics with the same settings used for analyzing the PGRPs and GNBPs. SignalP (version 6.0) and DeepTMHMM (version 1.0) were used to search for signal peptide sequences and transmembrane domains, respectively, in the CTLs [46,47].

## 3. Results

### 3.1. Construction of the Genome Sequences of M. separata

Using the long-read and short-read genome sequence data of *M. separata*, we constructed de novo genome sequences. The genome construction scheme is shown in Figure 1. The assembled genome sequence data were first constructed using only the long-read sequence data, with a mean coverage of 187×. Subsequently, the assembled genome sequence data were polished using the short-read sequence data. The basal status of the polished genome sequence data is shown in Table 1. The total length of the polished genome data, with a high contig N50 value (2,745,150 bp), is not unusual considering the genome sizes of other lepidoptera species. Subsequently, BUSCO was used to assess the completeness of the polished genome data (Table 2). Although most of the core genes in the BUSCO dataset were found in the polished genome data (Complete BUSCO), a few core genes, which are single-copy genes, were found to be duplicated. The reason for the duplications could be the high regional heterogeneity in diploid genomes, which could have eventually led to incorrect assembly of two contigs that were assembled as one contig [48]. Therefore, the polished genome data were loaded into the Purge Haplotigs to address this problem. The improved genome data from the Purge Haplotigs (referred to as the finished genome data in Table 2) were analyzed using BUSCO (Table 2). Almost all core genes in the eukaryota_odb10 or lepidoptera_odb10 datasets were found to be complete and single-copy BUSCO in the improved genome data, indicating that these genome data have good qualities as reference genome data with a CG percentage of 38.60%. Hereafter, we refer to these genome data as the reference genome data or simply “the genome data” (deposited in DDBJ/ENA/GenBank; see Section 2.2).

### 3.2. Search for Repetitive Region and Gene Prediction

We searched for the repetitive regions and transposable elements (TEs) in the genome sequence (Figure 1). First, de novo TE detection was performed and 1905 consensus sequences of TEs were constructed (Appendix A). TEs and repetitive regions in the *M. separata* genome were identified using consensus sequences; the summary status is shown in Table 3, and the output files are provided in Appendix A. Approximately 46.59% of the *M. separata* genome was repetitive or contained TE sites. Among the annotated TEs, the number of retroelements (Class I TEs) was much larger than that of DNA transposons (Class II TEs). Long interspersed nuclear elements (LINEs) were the most abundant retroelements, whereas Tc1-IS630-Pogo was the most abundant DNA transposon. Rolling circles covered 6.86% of the *M. separata* genome, whereas satellites and simple repeats were less than 1%.

Gene prediction of *M. separata* was performed using the masked genome data (Appendix A) and RNA-Seq data as hint data, of which a part were retrieved from SRA public database (Appendix A and Figure 1). A total of 21,970 genes were identified in the *M. separata* genome (gene.gtf in Appendix A), whereas 24,452 coding sequences (CDSs) and amino acid sequences were predicted because some genes had multiple transcripts (gene_cds.fasta and gene_pep.fasta in Appendix A). Using the predicted amino acid sequences, the predicted genes were functionally annotated with Fanflow4Insects (Appendix A) [43]. Approximately 45–57% of the protein sequences were annotated using gene sets of model species (*Homo sapiens*, *Mus musculus*, and *Caenorhabditis elegans* (Maupas, 1900)) (Table 4). Among the insect species datasets, a relatively higher percentage of protein sequences were annotated using Lepidoptera datasets (*M. sexta* and *B. mori*) than using other insect species datasets (*D. melanogaster*, the western honeybee (*Apis mellifera*; Hymenoptera; Linnaeus, 1758)*,* and the red flour beetle (*Tribolium castaneum*; Coleoptera; Herbst, 1797)), and over half of the sequences were annotated using the Pfam and Unigene datasets.

We checked whether the microbial sequences were contaminated in the reference genome and attempted to remove the microbial sequences using annotation data of the gene set with NCBI-nr (Appendix A). We counted the numbers of the names of the microbial species in the annotation data (Appendix A). A total of 32, 23, 39, 12, and 10 genes were annotated with orthologs from *Pseudomonas aeruginosa*, *Piscirickettsia salmonis*, Spodoptera moth adenovirus 1, *Trichoplusia ni* TED virus, and *Conidiobolus coronatus*, respectively. If these genes are located in the same contig and the contig sequences have sequences identical to those in the microbial species, the corresponding contigs need to be removed from the reference genome. The gene sets annotated with *P. aeruginosa*, *P. salmonis*, Spodoptera moth adenovirus 1, and Trichoplusia ni TED virus descriptions were not located in the same contigs. Ten genes annotated with the *C. coronatus* descriptions were located in the contig; however, this contig did not hit any genome sequences of *C. coronatus*. Therefore, we did not remove any contigs from the *M. separata* reference genome sequences.

### 3.3. Immune-Related Genes in M. Separata

To determine whether antimicrobial peptide (AMP) production systems function in *M. separata*, we searched for genes in the Toll and IMD pathways, which have been thoroughly investigated in *D. melanogaster* [19]. Using the datasets for the Toll and IMD pathway sequences in FlyBase, we determined whether the orthologs of the core components of the Toll and IMD pathways exist in *M. separata*. Orthologs of almost all core intracellular components of the Toll and IMD pathways were found in *M. separata* (Figure 2). Although *IMD* and *MyD88* were not found by BLASTx (Appendix A), both genes were identified by BLASTp (Appendix A).

Extracellular components in Toll pathways were searched [19]. Orthologues of *ModSP*, *Grass*, and *Spaezle-processing enzyme* (*SPE*), which form a serine protease cascade between sensor proteins and the Toll receptor complex, were found in *M. separata* genes, while the *Toll* (*Tl*) and *Spaezle* (*Spz*) (found by the BLASTp search) orthologs consisting of the Toll receptor complex were found as well (Figure 2). Sensor proteins in the IMD and Toll pathways recognize the surface structures of infected microbes and arouse signals to activate these pathways. PGRPs recognize the peptidoglycan structure derived from gram-positive or gram-negative bacteria. BLASTx results showed that nine genes in the *M. separata* genome were orthologs of PGRPs; orthologs of *PGRP-LE* and -*LC* (the IMD pathway) and *PGRP-SA* were found, and of the genes showing minimum e-values among *M. separata*, nine PGRPs in the BLASTx results were allocated as orthologs of each three PGRPs (Figure 2). GNBPs act as sensor proteins for fungi and gram-positive bacteria. Seven GNBP-encoding genes were found in the *M. separata* genome, whereas *D. melanogaster* harbored only three such genes. Orthologs of GNBP1 and GNBP3 showing minimum e-values among *M. separata* GNBPs in the BLASTx results were allocated (Figure 2). Furthermore, the NJ phylogenetic trees of PGRPs and GNBPs were constructed, domain analysis was performed by hmmerscan bundled in DoMosaics, and the ML phylogenetic trees of PGRP and GNBP were constructed (Appendix A). In the PGRP NJ tree, three *D. melanogaster* PGRPs and *M. separata* anno1.g18688.t1 (annotated PGRP-LC and LE by BLASTx results) formed a clade, while *M. separata* anno1.g9300.t1 belonged to another clade. The domain analysis of the PGRP showed that most of *M. separata* PGRPs possessed the Amidase_2 domain, which *D. melanogaster* PGRPs harbored. The structure of the ML tree of PGRP was different from that of the NJ tree of PGRP. In the ML tree, *D. melanogaster* PGRP-LC and *D. melanogaster* PGRP-SA formed a clade with anno1.g9298.t and anno1.g14545.t, respectively, while *D. melanogaster* PGRP-LE did not form a clade with a single *M. separata* PGRP. In the GNBP NJ tree, anno1.g3877.t1 and anno1.g3876 formed the same clades of *D. melanogaster* GNBP1 and GNBP3, respectively, and both two *M. separata* GNBPs possessed the CBM39 and Glyco_hydro_16 domains, which *D. melanogaster* GNBPs harbored. The structures of the ML tree were different from that of the NJ tree. *D. melanogaster* GNBP1 formed a clade with two *M. separata* GNBPs (anno1.g20324 and anno1.g20324), which are considered as paralogous genes, while *D. melanogaster* GNBP3 did not form a clade with a single *M. separata* GNBP. Taken together, there are different relationships between the orthologs of *D. melanogaster* and *M. separata* in the NJ and ML trees. Furthermore, there are inconsistencies between the BLASTx and phylogenetic tree results. The reasons for this might be because the methods of calculations of the sequence similarities were different, and the degrees of differences within PGRPs and GNBPs sequences might not be as high, especially in PGRPs, due to low bootstrap values in the ML tree. Functional analysis of these proteins of *M. separata* is required in future research.

AMPs are effector molecules against invading microbes; in *D. melanogaster*, the induction of AMP genes is regulated mainly by the Toll and IMD pathways [49]. Defensin, cecropin, attacin, gloverin, moricin, and lebocin are the major AMP genes in insects [50]. Using the *M. separata* gene dataset and annotation results (Appendix A), orthologs of the AMP genes in *M. separata* were searched (Appendix A). This is because most AMPs are very short, and orthologs cannot be found through BLAST alone. Although orthologs of defensin and cecropin were not found with low E-values by either BLASTp or hmmer domain search (data not shown), eleven defensins and ten cecropins were found in the annotation data of *M. separata*. The eleven defensins were annotated with “Defensin” descriptions of several insect species such as *D. melanogaster* and *Aedes aegypti* (“UniGene-description” column in Appendix A), whereas the ten cecropins were annotated with “Cecropin” descriptions of *M. sexta*. Six attacins, five gloverins (anno2.g16145.t1 was also annotated as attacin), nine moricins (with seven moricins were also annotated as cecropins), and five lebocins were found in *M. separata*.

As described above, CTLs recognize a wide range of ligands [51,52]. Therefore, CTLs are involved in several immune reactions of insects by binding to the surfaces of both the invaded microbes and organisms and their own cells and tissues. CTLs possess more than one C-type lectin domain (CTLD) (known as the “carbohydrate recognition domains”, or CRD). Thus, CTL genes in *M. separata,* which were annotated as possessing CTLD (Pfam ID: PF00059)*,* were searched. Among *M. separata* genes, 105 CTLs were identified (Appendix A). The results of the domain analysis by hmmerscan bundled in DoMosaics and sequences analysis of the 105 CTL are shown in Appendix A. Most CTLs possessed two CTLDs, and were classified as Dual CTLDs, while twelve possessed single CTLDs and six possessed more than three CTLDs (Table 5). Five CTLs possessed CTLD and other domains; these were classified as CTL-X. Furthermore, we determined whether the CTLs possessed signal peptide sequences and transmembrane domains. Of the 105 CTLs, 77 were predicted to possess signal peptides using SignalP and 84 were predicted to possess signal peptides using DeepTMHMM (Table 5). This difference in the number of predicted signal peptides may be because of the difference in the prediction algorithms of the two software types we used. These four CTLs all possessed a transmembrane domain.

## 4. Discussion

In this study, we constructed the reference genome sequences and gene dataset of the oriental armyworm, *M. separata*. Using long-read and short-read genome data, genome sequences with high contig N50 values and good BUSCO scores were constructed through polishing and purging haplotig processes. After searching for repetitive regions and TEs in the genome sequences and masking such sequences, we obtained gene set data using masked genome data and RNA-Seq data, some of which we retrieved from SRA. Consequently, the gene set was functionally annotated; using amino acid sequence data of the gene set and the functional annotation data, orthologs of the IMD and Toll pathways were then identified, as well as CTLs, which are involved in immune reactions in model insect species.

The contig N50 value of the reference genome we constructed was approximated at 2.7 Mbp, and the results of the BUSCO analysis showed that over 98% of BUSCO genes (core genes) were identified. These results suggest that the genome sequence data of *M. separata* possess sufficient quality for use as a reference genome in term of the continuity and completeness of genome, and can be used for the various studies, especially genomic or molecular research. The size of the finished *M. separata* genome sequence was approximately 628 Mbp, as against those of *P. xylostella, T. ni*, *B. mori*, *S. litura*, *S. exigua*, and *M. sexta*, which were 328 Mbp, 333 Mbp, 460 Mbp, 438 Mbp, 419 Mbp, and 470 Mbp, respectively [2,9,12,13,15,17]. The genome size of butterflies (264 species) varied from 195 to 1262 Mbp [18]. Considering these results, the genome size of *M. separata* is larger than those of the other many lepidopteran species, with the exception of butterflies, and is within a reasonable range of other lepidopteran insects.

Further, it was found that approximately 48% of the *M. separata* genome contained repetitive regions or TEs; of these, retroelements were the major TEs, of which LINEs were the major components. In *B. mori*, approximately 46.5% of the genome contained repetitive regions or TEs, and more Class I retroelements than Class II DNA elements were detected, which is the same tendency as in the *M. separata* genome [9]. Additionally, SINEs consist of approximately 12% of the *B. mori* genome, and LINEs consist of approximately 17%, which are comparable numbers. However, approximately 32% of the *S. litura* genome and more Class I retroelements (about 11%) than Class II DNA elements (about 2%) were detected. LINEs and SINEs consist of about 8% and 2% of the *S. litura* genome, respectively [2]. In *A. mellifera*, a hymenopteran model species, the total repetitive region consists of approximately 11% of the genome, and Class II DNA elements are major components of the TEs, which are clearly different from lepidopteran families [53]. Taken together, these results suggest that the comprehensive tendency of *M. separata* is the same as the model lepidopteran species, while detailed status, especially the ratio of SINEs, is different, and may represent *M. separata*-specific features of TEs.

A total of 21,970 genes (24,452 CDSs) were predicted in the *M. separata* genome, whereas *B. mori*, *S. litura*, *S. exigua*, and *M. sexta* had 16,880 genes, 15,317 protein-coding genes, 18,477 transcripts, and 25,256 genes, respectively [2,9,12,17]. The differences among species may be partially owing to the methods used to prepare the gene sets. Considering these results, the number of genes of *M. separata* may be considered reasonable for a lepidopteran gene set. The annotation ratios of *H. sapiens*, *M. musculus*, and *C. elegans* are lower than those of *T. castaneum*, *M. sexta*, and *B. mori.* The ratio of *D. melanogaster*, a model insect species, to *A. mellifera* is not very high. According to a phylogenetic tree constructed by Misof et al. [54], Hymenoptera branched at the earliest time, then Coleoptera branched, and then Lepidoptera and Diptera branched. Our ratios were not consistent with this tree structure. Although we could not determine the underlying reasons for these differences, the gene sets could provide new evolutionary insights in such research fields.

Searching for the genes consisting of the Toll and IMD pathways revealed that intracellular components of these pathways are conserved, which has previously been observed in multiple insect species [19,55,56,57], suggesting that AMPs are regulated by the two signaling pathways in *M. separata*. The numbers of PGRPs and GNBPs predicted to function as pattern recognition receptors (PRRs) for fungi and bacteria [58] were different from those in other insect species [19,55,57,59,60]. Nine PGRPs and seven GNBPs were identified in the *M. separata* genome. Among these genes, the orthologs of PGRP-LC, -LE, and -SA, GNBP1, and GNBP3 function as PRRs in *D. melanogaster*. However, functional analysis using non-*Drosophila* species showed that the orthologs did not play a pivotal role in sensing microbes [60,61], which led to different signal transduction in the Toll and IMD pathways in *D. melanogaster*. These results suggest that *M. separata* has the Toll and IMD pathway systems, which have different signal transduction flows from *D. melanogaster*, as observed in *Plautia stali* and *T. castaneum* [60,62]. Over forty AMP genes were identified in the *M. separata* genome, including typical AMP genes such as cecropin, defensin, and attacin. However, several of the AMP genes were annotated without typical domain annotations in the Pfam database analysis. For example, the cecropin domain was not identified in the ten cecropin genes annotated by the *M. sexta* gene set. Thus, further investigation using RNA-Seq or quantitative reverse transcription-PCR analysis is warranted in order to determine whether such AMP genes actually exist or are expressed.

According to Xia et al., the number of CTL genes in insects ranges from four to forty [52]. Among lepidoptera species, *B. mori* and *M. sexta* have 23 and 34 CTL genes, respectively, whereas *P. xylostella* has only seven genes. Compared to these species, *M. separata* has an outstandingly higher number of CTL genes. Notably, approximately 90% of CTLs have two CTLDs (or CRDs), which we categorized as dual CTLD-type CTLs and more CTLD-type CTLs. Additionally, approximately >70% CTLs were found to have a signal peptide sequence, five CTLs were classified as CTL-X, and four CTLs had transmembrane domains; these results are similar to those obtained for other insects [52]. Based on these results, we assumed that the dual CTLD-type lectin genes were duplicated in the *M. separata* genome. Many dual CTLD-type lectins regulate the immune system of insects. Immunlectin II, a dual CTLD-type lectin from *M. sexta*, binds to bacterial lipopolysaccharide with a CTLD [63]. In *B. mori*, dual CTLD-type CTLs are required for nodule formation [64]. EPL, which enhances encapsulation, is a dual CTLD-type lectin [21]. Therefore, dual CTLD-type lectins are expected to play more important roles in immunity in *M. separata* than in other insects. Recently, Sawa et al. reported that the parasitoid wasp *Cotesia cariyai* suppressed melanization and encapsulation of *M. separata* by manipulating the CTLs of both species, suggesting that CTLs are key factors for the success or failure of parasitization [65]. To shed light on the well-controlled immune system and its interaction with various microorganisms and parasitoids, analyzing the diversified lectins and the molecules they recognize is necessary.

## 5. Conclusions

We constructed a reference genome of *M. separata* (size: 682 Mbp), with the final genome having sufficient quality as a reference genome in terms of both continuity and completeness of core genes. In addition, 21,970 genes were predicted using genome sequence data, and functional annotations were performed. Using the gene set and annotation data, several immune-related genes were identified, providing new insights into the genomics and immunology of *M. separata*. We believe that the obtained reference genome data can promote studies on molecular biology of *M. separata* and comparative genomics of insects in general.

## Figures and Tables

**Figure 1 insects-13-01172-f001:**
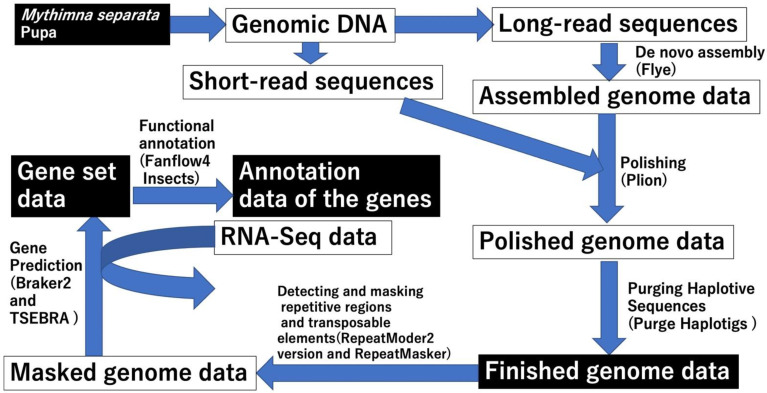
Schemes of reference genome construction, gene prediction, and functional annotating. “Finished genome data” indicates the reference genome data. The names of software used in each step are shown in brackets.

**Figure 2 insects-13-01172-f002:**
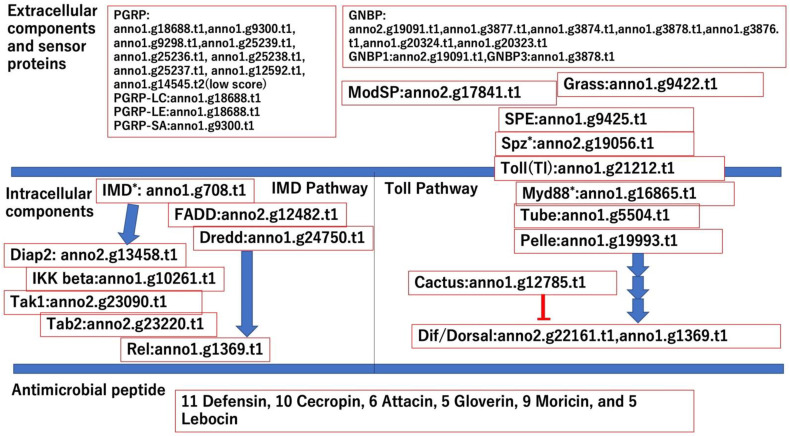
Gene IDs of the orthologues consisting of Toll and IMD pathway based on BLASTx search. The number of genes of antimicrobial peptides is shown in bottom of this figure, and the gene IDs of the orthologs of antimicrobial peptides are shown in Appendix A. Asterisks indicate that the orthologs were found by BLASTp search.

**Table 1 insects-13-01172-t001:** The status of the constructed genome sequence data *.

Genome Sequence Name	Polished Genome Data	Finished Genome Data (Reference Genome Data)
Total length (bp)	842,875,911	681,943,315
Contig number	991	569
Contig N50 (bp)	2,220,816	2,745,150
Largest contig length (bp)	12,409,004	12,409,004
Average of contig length (bp)	850,530.69	1,198,494.40

* Both data contain no N sequence and gap.

**Table 2 insects-13-01172-t002:** The results of BUSCO using the constructed genome sequence data.

Genome Sequence Name (BUSCO Data Set)	Polished Genome Data (eukaryota_odb10)	Finished Genome Data (eukaryota_odb10)	Finished Genome Data (lepidoptera_odb10)
Complete BUSCOs	253 (99.2%)	252 (98.8%)	5213 (98.6%)
Complete and single-copy BUSCOs	178 (69.8%)	251 (98.4%)	5185 (98.1%)
Complete and duplicated BUSCOs	75 (29.4%)	1 (0.4%)	28 (0.5%)
Fragmented BUSCOs	1 (0.4%)	2 (0.8%)	20 (0.4%)
Missing BUSCOs	1 (0.4%)	1 (0.4%)	53 (1.0%)
Total BUSCO groups searched	255	255	5286

**Table 3 insects-13-01172-t003:** Transposons and repetitive regions in *M. separata* genome.

Total Length:	681,943,315 bp		
Bases Masked:	317,706,294 bp (46.59%)		
	Number of Elements *	Length Occupied	Percentage of Sequence
Retroelements	483,943	102,500,601 bp	15.03%
SINEs:	69	3659 bp	0.00%
Penelope	0	0 bp	0.00%
LINEs:	466,443	94,162,645 bp	13.81%
CRE/SLACS	11,265	2,482,075 bp	0.36%
L2/CR1/Rex	47,274	15,582,975 bp	2.29%
R1/LOA/Jockey	125,283	28,136,850 bp	4.13%
R2/R4/NeSL	8079	2,985,258 bp	0.44%
RTE/Bov-B	184,015	31,548,956 bp	4.63%
L1/CIN4	0	0 bp	0.00%
LTR elements:	17,431	8,334,297 bp	1.22%
BEL/Pao	1963	3,457,738 bp	0.51%
Ty1/Copia	2715	1,358,093 bp	0.20%
Gypsy/DIRS1	6472	2,769,678 bp	0.41%
Retroviral	160	76,839 bp	0.01%
DNA transposons	80,267	20,701,313 bp	3.04%
hobo-Activator	6121	1,085,735 bp	0.16%
Tc1-IS630-Pogo	24,800	10,838,060 bp	1.59%
En-Spm	0	0 bp	0.00%
MuDR-IS905	0	0 bp	0.00%
PiggyBac	166	236,668 bp	0.03%
Tourist/Harbinger	2055	550,091 bp	0.08%
Other (Mirage, P-element, Transib)	110	177,033 bp	0.03%
Rolling-circles	281,101	46,800,852 bp	6.86%
Unclassified:	880,600	140,919,871 bp	20.66%
Total interspersed repeats:		264,121,785 bp	38.73%
Small RNA:	204	23,041 bp	0.00%
Satellites:	177	69,651 bp	0.01%
Simple repeats:	102,349	5,964,291 bp	0.87%
Low complexity:	15,392	726,674 bp	0.11%

* most repeats fragmented by insertions or deletions have been counted as one element.

**Table 4 insects-13-01172-t004:** Numbers of hit genes in each data set by Fanflow4.

	Numbers of the Hit Proteins (among 24,453 Proteins)	Percentage of the Hit Proteins (%)
*H. sapience*	14,037	57.4039995
*M. musculus*	13,699	56.021756
*C. elegans*	11,235	45.9452828
*D. melanogaster*	12,879	52.6683842
*B. mori*	19,704	80.5790701
*M. sexta*	17,996	73.594242
*A. mellifera*	14,427	58.9988958
*T. castaneum*	16,496	67.4600254
Unigene	16,086	65.7833395
Pfam	13,244	54.1610436

**Table 5 insects-13-01172-t005:** Summary of CTL analyses.

Total Number of CTLs	Single CTLD *	Dual CTLD (Immune-Lectin Group or Dual-CTLD Type Lectins) *	More than Three CTLD	CTL-X Group *	Signal Peptide (SignalP)	Signal Peptide (DeepTMHMM)	Transmembrane Domain
105	12	81	7	5	77	84	4

* Classifications are based on Xia et al., 2018 [52].

## Data Availability

All Appendix A are available in figshare. DOI: 10.6084/m9.figshare.c.6192784.

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
