# Peer review of "Reference Genome Sequences of the Oriental Armyworm, Mythimna separata (Lepidoptera: Noctuidae)"

_insects, 2022, doi:10.3390/insects13121172_

Round 1

Reviewer 1 Report

In this manuscript, Yokoi and colleagues sequenced and assembled a reference genome of armyworm Mythimna separata, an agricultural pest. The methods and analyses carried out in this study is standard and straightforward. I have the following comments and suggestions, which hopefully can improve the manuscript before its publication.

Required

1) For the extracellular components and sensor proteins, intracellular components, and Toll pathway genes, in addition to using Blast search, the authors should construct ML and BI gene trees to confidently assign their identities.

2) The supplementary information was not provided for me to review. Judging from the legends, they are the raw sequencing data, assembled genome, predicted proteins, etc. I just want to kindly remind the authors that they should be made available to the publication of this manuscript.

Optional

1) Figure 1 - The authors may consider also adding in the software used for each step.

2) Table 3 - It would be nice if the authors may consider put comma for the numbers as they have done in Table 1. For example, instead of 102500601 bp, showing it in format of 102,500,601 bp, will be easier for readers to read.

3) I understand that the authors of the this manuscript are probably experts familiar with the different species names. A suggestion for the authors for consideration is to add the common name for all species throughout the whole manuscript, such that general readers can also get a better understanding. For example, in the title, armyworm can be added. 

Typos

- line 41 and 297, 2.7Mbp instead of 2.7M

- line 166, 187X instead of 187

- for the N50, I presume the authors are referring them as scaffold N50. Please kindly revise throughout the manuscript for clarity.

Reviewer 2 Report

This is an interesting and nice paper.

 I suggest you check the article with someone whose mother tongue is English (mine is not).

All Latin names of taxa presented for the first time in the text should be accompanied by the author’s name and the description date.

I think that, you used the phrase „lepidopteran insects” too many times (22). I suggest that you sometimes use the „Lepidoptera species” „lepidopteran species”, „lepidopteran families” (or you can use the „moths” to Noctuidae species)

I suggest that you sometimes use the common english name (oriental armyworm) instead of the sientific name of the species.

Other specific suggestions and corrections:

line 19 delete hyphen between 682 and Mbp

line 168 add the N50 value (xx)

line 196 Delete the A before Among

line 239 misspelling (dot)

line 255 I think “which act as sensor proteins for gram-positive bacteria and fungi” is unnecessary to write again, because you wrote about it in line 253

line 265 insert the citations

line 306 lepidopteran
